

# A citizen science approach to monitoring bleaching in the zoantharian *Palythoa tuberculosa*

John Everett Parkinson[1], Sung-Yin Yang[1,2], Iori Kawamura[1], Gordon Byron[1,3], Peter Alan Todd[4] and James Davis Reimer[1,5]

[1] Molecular Invertebrate Systematics and Ecology Laboratory, Department of Chemistry, Biology, and Marine Science, Faculty of Science, University of the Ryukyus, Nishihara, Okinawa, Japan
[2] Microbiology and Biochemistry of Secondary Metabolites Unit, Okinawa Institute of Science and Technology Graduate University, Onna, Okinawa, Japan
[3] Red Sea Research Center, Division of Biological and Environmental Science and Engineering, King Abdullah University of Science and Technology (KAUST), Thuwal, Saudi Arabia
[4] Experimental Marine Ecology Laboratory, National University of Singapore, Republic of Singapore
[5] Tropical Biosphere Research Center, University of the Ryukyus, Nishihara, Okinawa, Japan

Corresponding author
John Everett Parkinson,
jparkinson@psu.edu

## ABSTRACT

Coral reef bleaching events are expected to become more frequent and severe in the near future as climate changes. The zoantharian *Palythoa tuberculosa* bleaches earlier than many scleractinian corals and may serve as an indicator species. Basic monitoring of such species could help to detect and even anticipate bleaching events, especially in areas where more sophisticated approaches that rely on buoy or satellite measurements of sea surface temperature are unavailable or too coarse. One simple and inexpensive monitoring method involves training volunteers to record observations of host color as a proxy for symbiosis quality. Here, we trained university students to take the 'color fingerprint' of a reef by assessing the color of multiple randomly selected colonies of *P. tuberculosa* at one time point in Okinawa Island, Japan. We tested the reliability of the students' color scores and whether they matched expectations based on previous monthly monitoring of tagged colonies at the same locations. We also measured three traditional metrics of symbiosis quality for comparison: symbiont morphological condition, cell density, and chlorophyll *a* content. We found that *P. tuberculosa* color score, although highly correlated among observers, provided little predictive power for the other variables. This was likely due to inherent variation in colony color among generally healthy zoantharians in midwinter, as well as low sample size and brief training owing to the course structure. Despite certain limitations of *P. tuberculosa* as a focal organism, the citizen science approach to color monitoring has promise, and we outline steps that could improve similar efforts in the future.

Subjects Conservation Biology, Marine Biology
Keywords Bleaching, Citizen science, Chlorophyll, Color, Coral, Zoantharian, Zooxanthellae

## INTRODUCTION

Visible loss of color among zooxanthellate cnidarian colonies can reflect a stress-induced breakdown of the mutualistic relationship between hosts and their endosymbiotic algae, dinoflagellates of the genus *Symbiodinium* Freudenthal. As the association deteriorates,

reductions in photosynthetic pigment levels and/or healthy *Symbiodinium* densities dampen the color intensity of the colony (*Fitt et al., 2001*). Bleaching is driven by many factors, chief among them increasing sea surface temperatures associated with ongoing climate change (*Brown, 1997*). Mass bleaching events have been observed in almost all subtropical and tropical reef areas (*Wilkinson, 1998*; *Goldberg & Wilkinson, 2004*; *Donner et al., 2005*), sometimes causing extensive mortality of hard corals (e.g. *Glynn et al., 2001*; *Loya et al., 2001*; *Depczynski et al., 2013*), soft corals (e.g., *Harvell et al., 2001*; *Prada, Weil & Yoshioka, 2010*; *Dias & Gondim, 2015*), and zoantharians (e.g., *Williams Jr & Bunkley-Williams, 1990*; *Goldberg & Wilkinson, 2004*). Bleaching is one of the most serious problems facing the long-term survival of coral reef ecosystems (*Hoegh-Guldberg, 1999*), especially as models predict bleaching to become an annual event for many reefs in the near future (*Donner et al., 2005*; *Van Hooidonk, Maynard & Planes, 2013*). Thus, detecting and predicting bleaching events on broad and local scales has become a key component of monitoring the health of coral reef ecosystems (*Andréfouët et al., 2002*; *Eakin, Lough & Heron, 2009*).

Although buoy and satellite-based products have greatly improved our capacity to anticipate bleaching events (*Eakin, Lough & Heron, 2009*), the data are not always readily available in certain areas, and local variability such as infrequent upwelling or freshwater input may necessitate targeted observation. For many managers monitoring reefs at the local scale, it is often advantageous to take a citizen science approach, and to adopt simplified tools and techniques that are easily taught to volunteers (*Hunter, Alabri & Ingen, 2013*). One example is the 'CoralWatch Coral Health Chart' (*Siebeck et al., 2006*), a waterproof card featuring standardized color squares for determining changes in bleaching status. In practice, observers compare the reference card and a host colony *in situ,* scoring colors on a saturation scale ranging from one to six (*Siebeck et al., 2006*; *Siebeck, Logan & Marshall, 2008*). These scores correlate with chlorophyll *a* concentrations and *Symbiodinium* densities in hard corals (*Siebeck et al., 2006*) and to some degree with the morphological condition of *Symbiodinium* in zoantharians (*Hibino et al., 2013*). The method can scale to track colonies, species, or communities, and has been used to monitor reef symbiosis quality over time (*Cooper, Gilmour & Fabricius, 2009*; *Montano et al., 2010*; *Marshall, Kleine & Dean, 2012*; *Paley & Bay, 2012*; *Tanzil, 2012*).

Color cards are also recommended for taking a 'fingerprint' of the reef condition, which involves observing a random cross-section of colonies at a single time point (*Siebeck, Logan & Marshall, 2008*). Not to be confused with other techniques like DNA fingerprinting, 'color fingerprinting' is an alternative to repeatedly observing tagged colonies ('recurrent monitoring'). Although less precise than recurrent monitoring, color fingerprinting does have the power to detect differences within and among bleaching and recovered reefs, and is more suitable for non-specialists than recurrent monitoring because the method does not require a permanent transect or tagged colonies (*Siebeck, Logan & Marshall, 2008*). Certain anthozoans such as *Acropora, Millepora*, and *Palythoa* in particular tend to visibly bleach sooner than other members of the reef community (*Williams Jr & Bunkley-Williams, 1990*; *Wilkinson, 1998*). These 'early-indicator' species might serve as useful color fingerprinting targets to anticipate changes in bleaching condition at a given site.

**Parkinson et al. (2016),** *PeerJ,* **DOI 10.7717/peerj.1815**                                                                 **2/14**

The zoantharian *Palythoa tuberculosa* (Esper) is broadly distributed throughout the Indo-Pacific, from Madagascar to the Red Sea to Japan (*Hibino et al., 2014*; *Reimer et al., 2007*). It is abundant in shallow waters, easily observed via snorkeling, and free from legal protections that limit sampling of other cnidarians, making it a practical candidate for observation. It has been suggested that certain Caribbean *Palythoa* species should not be used as indicators because their "bleaching" has not correlated with temperature or season and may have simply reflected natural color variation (*Cook et al., 1990*). Moreover, many zoantharians incorporate calcareous sand into their bodies, potentially affecting color (e.g., *West, 1979*). However, in 2009 *Hibino et al. (2013)* monitored 12 tagged colonies of *P. tuberculosa* monthly in Okinawa Island, Japan, and found that many colonies paled during periods of rapidly elevating sea surface temperature (e.g., from 21 °C to 25 °C in 3 weeks) and that colony color correlated with symbiont morphological condition. Thus, continued *P. tuberculosa* color observation at this location seems warranted, though in need of additional testing.

Here, our goal was to further develop methodologies for monitoring zoantharian bleaching condition by incorporating citizen scientists in color score assessment and testing the accuracy and utility of their data. In January 2014, we trained 12 student volunteers during a two-day course in the use of the CoralWatch Coral Health Chart. The students scored 20 new colonies of *P. tuberculosa* from Okinawa Island at the same locations described by *Hibino et al. (2013)*. To determine whether the data acquired by naïve observers were reliable, we compared color scores among the students, who were trained together but worked independently on the same colony images. To assess whether color corresponded to symbiosis quality, we also measured three alternative metrics: symbiont morphological condition, cell density, and chlorophyll *a* content. Finally, we placed the new color fingerprinting data in context by comparing them to the previously published recurrent monitoring time series data.

## MATERIALS & METHODS

### Training for color card scoring and *Symbiodinium* morphological condition observation

Twelve undergraduate students from the University of the Ryukyus were trained as part of a two-day practical course using images of *Palythoa tuberculosa* and *Symbiodinium* sp. collected previously as part of a year-long monitoring project (*Hibino et al., 2013*). During lecture, students were shown 20 images of *P. tuberculosa* colonies and asked to independently score colony colors with reference to 'CoralWatch Coral Health Chart' color cards included in the photographs. Color was scored from a low of 0.0 (completely white) to a high of 6.0 (very dark) by increments of 0.5 (as by *Hibino et al., 2013*), effectively doubling the resolution of the original scale (*Siebeck et al., 2006*). Additionally, students were shown 20 slides of *Symbiodinium* cells collected from the same colonies and asked to independently classify *Symbiodinium* morphological conditions. Each student calculated the percentage of 'normal' (non-degraded) zooxanthellae (NZ%) according to *Reimer et al. (2007)*. The instructor checked student color scores and NZ% values against those of

*Hibino et al. (2013)* to confirm accuracy. Because deviations were generally low, students were deemed competent for field work.

## Colony collection, image capture, and color scoring

Two locations in Okinawa, Japan, were included in our color fingerprinting assessment and matched those sampled by *Hibino et al. (2013)*: Odo Beach (26°05′N, 127°42′E) in Itoman City, and Mizugama (26°21′N, 127°44′E), in Kadena Town, very close to Miyagi Beach (for a map, see Fig. 1 of *Hibino et al., 2013*). Colonies were sampled in the early afternoon of 12 Jan. 2014, when the intertidal zones of Odo and Mizugama were submerged. At both locations, 10 colonies of *P. tuberculosa* were selected haphazardly from the intertidal zone. All colonies were >50 cm$^2$, separated by >5 m, and positioned at depths <2 m.

Digital images of each colony were taken *in situ* with an underwater camera. A laminated 'CoralWatch Health Chart' color card was placed by each colony and included in the photograph, allowing for standard comparisons regardless of variable light levels or camera equipment. Color scores were later assessed independently by all 12 trained observers from the digital images. Because *P. tuberculosa* colonies occasionally possess some darker and lighter areas, the score was based on the most typical intermediate shade accounting for most of the colony's area. Immediately after the photographs were taken, fragments (~5 cm$^3$) of each colony were sampled with a diving knife and placed into individually marked sealable plastic bags. The bags were then placed in a 15 L bucket filled with ambient seawater and taken back to the laboratory at the University of the Ryukyus. Total transit time was <1 h.

## *Symbiodinium* morphological condition, cell density, and chlorophyll *a* content

The freshly collected *P. tuberculosa* fragments were further processed in the laboratory. *Symbiodinium* cell preparation followed *Hibino et al. (2013)*. Briefly, host tissue was macerated, homogenized, resuspended in 1 ml seawater, and spread on a hemocytometer (Thoma EKDS, 1/10 mm depth). *Symbiodinium* cells were visualized at 400× magnification with a light microscope and photographed ($n > 100$ cells per specimen). Cells were then tallied from the images according to the morphological condition classification scheme described by *Reimer et al. (2007)* based on previous studies (*Kuroki & Van Woesik, 1999*; *Mise & Hidaka, 2003*). Each student performed his or her own assessment to calculate NZ%.

Additional tissue from the *P. tuberculosa* fragments (1 cm$^3$ each) was used to calculate *Symbiodinium* cell densities. The tissue was processed and visualized as above, with the added step of using fine calipers to cut exact tissue dimensions so that relative densities would be directly comparable. Cell density per colony was calculated as the total number of cells observed divided by the total volume observed. Again, each student performed his or her own assessment.

To determine symbiont-derived chlorophyll *a* content in each *P. tuberculosa* colony, *Symbiodinium* cells were isolated following *Richier et al. (2003)* and *Wang et al. (2011)*. Briefly, calipers were used to cut 1 cm$^3$ of tissue which was then macerated, homogenized,

allowed to settle, cleaned through serial resuspension in seawater, and finally resuspended in 2 ml of 90% acetone and preserved at 4 °C overnight. The next day, chlorophyll *a* measurement followed *Jeffrey & Humphrey (1975)* and was performed on a Shimadzu UV-1800 UV spectrophotometer with absorbances at 630, 647, and 664 nm. Chlorophyll *a* content was standardized to the volume of the *P. tuberculosa* sample. Only one measurement was made per colony by an instructor; students did not contribute to this analysis.

## Statistical analyses

All analyses were performed in the R statistical environment. The original data files and R script can be accessed in Data S1. Statistical significance was assessed at $\alpha < 0.05$. For each colony in the study, color score was averaged from the values recorded by all 12 student observers, and coefficients of variation (standard deviation/mean) were calculated to estimate among-observer error. Reliability and correlation among student color scores were assessed with Krippendorf's alpha and Spearman's rho coefficients, respectively. Confidence intervals were generated via jackknife.

To investigate associations between color scores and other metrics, Spearman's rho coefficients were calculated with symbiont morphological condition (NZ%), cell density, or chlorophyll *a* content. Spearman's rho was used instead of Pearson's rho because the color scale is not continuous. For these calculations, the average color score per colony was rounded to the nearest 0.5 (or, if exactly intermediate, rounded down). For symbiont morphological condition and cell density, the unrounded average was used. For chlorophyll *a* content, the single measured value was used.

For comparative purposes, we reanalyzed the previously published time series data (*Hibino et al., 2013*; with permission). This earlier monitoring work was performed at or near the same two Okinawan locations and focused on the same host species, *P. tuberculosa*. It includes monthly color score and NZ% data collected from 12 tagged colonies throughout 2009. Correlations between raw color score and NZ% were again calculated with Spearman's rho.

## RESULTS

### Color score variation

*Palythoa tuberculosa* colonies at Odo Beach and Mizugama exhibited a moderate degree of color variation among different individuals (Fig. 1). Average colony color scores in this study ranged from 3.50 to 5.63, corresponding to a rounded range of 3.5 to 5.5. This span is typically associated with healthy colonies (*Siebeck et al., 2006*; *Siebeck, Logan & Marshall, 2008*). Agreement among observers was generally high. Most scores fell within ±1.0 of the rounded observer average (Fig. 2A). Accordingly, associated coefficients of variation were small relative to the interval size of 0.5 (mean CV: 0.10 ± 0.02 s.d.). However, for one colony (O7), the color score ranged from a low of 3.0 to a high of 6.0, spanning 7 intervals.

Krippendorf's alpha is a metric of agreement among observers scoring identical subjects on a common scale. In the absence of systemic bias, it ranges from 0 (no reliability) to 1 (perfect reliability), and values <0.667 are considered unreliable (*Krippendorff, 2004*). Observed alpha fell below this threshold (mean $\alpha = 0.555$, 95% CI [0.534–0.574]).

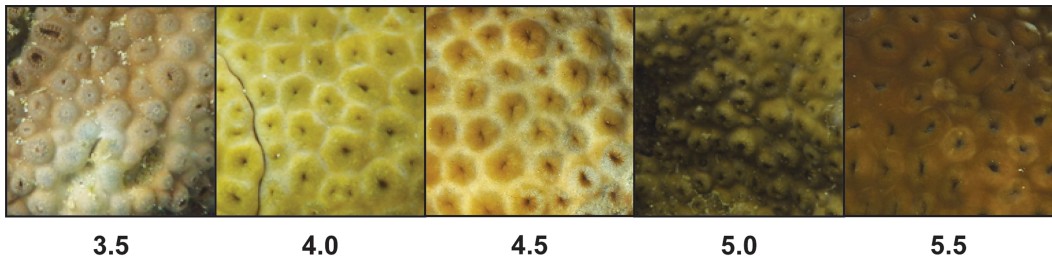

| 3.5 | 4.0 | 4.5 | 5.0 | 5.5 |

**Figure 1** **Color variation among unique colonies of *Palythoa tuberculosa* from reefs in Okinawa Island, Japan.** Colonies were photographed in January 2014 and standardized to the CoralWatch Coral Health Chart present in each image.

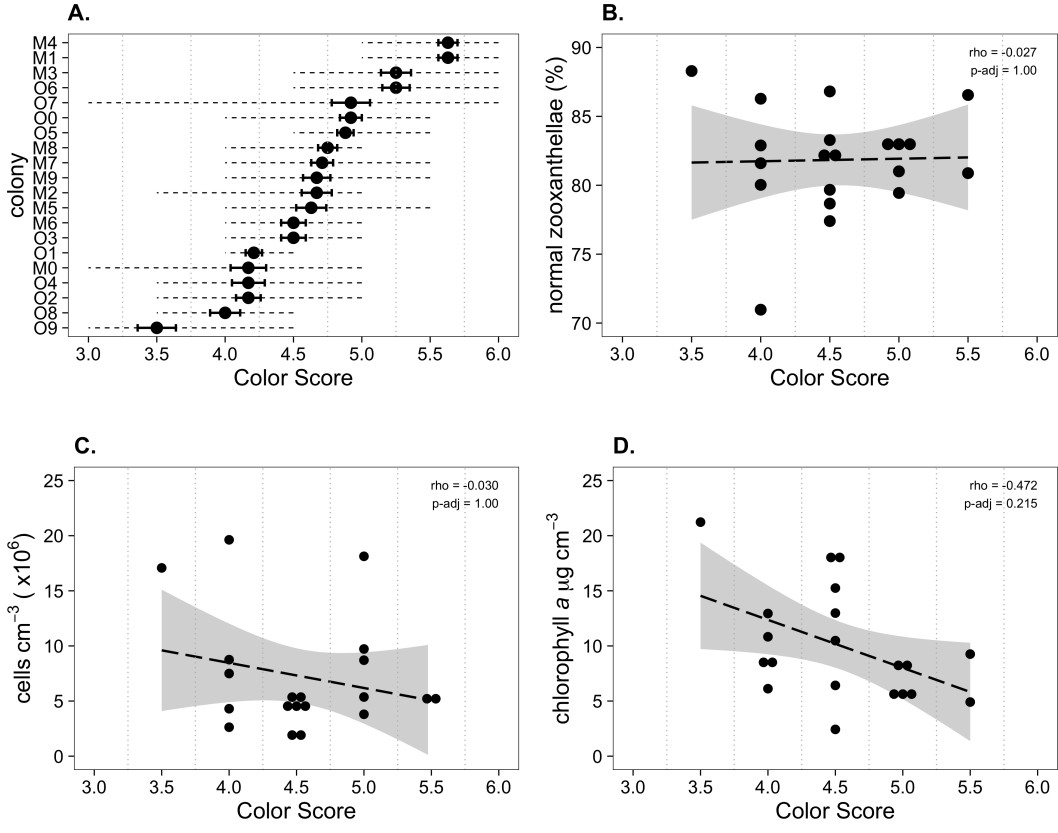

**Figure 2** **Physiological characteristics of *Palythoa tuberculosa* colonies in January 2014.** (A) Colony color score mean (dot), coefficient of variation (solid error bars), and range (dotted lines). O, Odo Beach; M, Mizugama. (B) 'Normal zooxanthellae' percent, (C) symbiont cell density, and (D) chlorophyll *a* content per colony segregated by color score. Dashed black lines indicate linear regression fits. Gray shading indicates 95% confidence intervals. Rho values are Spearman correlation coefficients. Significance was assessed at $\alpha = 0.05$.

Spearman's rho is a nonparametric measure of correlation that characterizes the extent to which two variables are described by a monotonic function. It ranges from $-1$ (perfect negative correlation) to 0 (no correlation) to $+1$ (perfect positive correlation). Observed rho among observers was positive (mean $\rho = 0.633$, 95% CI [0.590–0.661]).

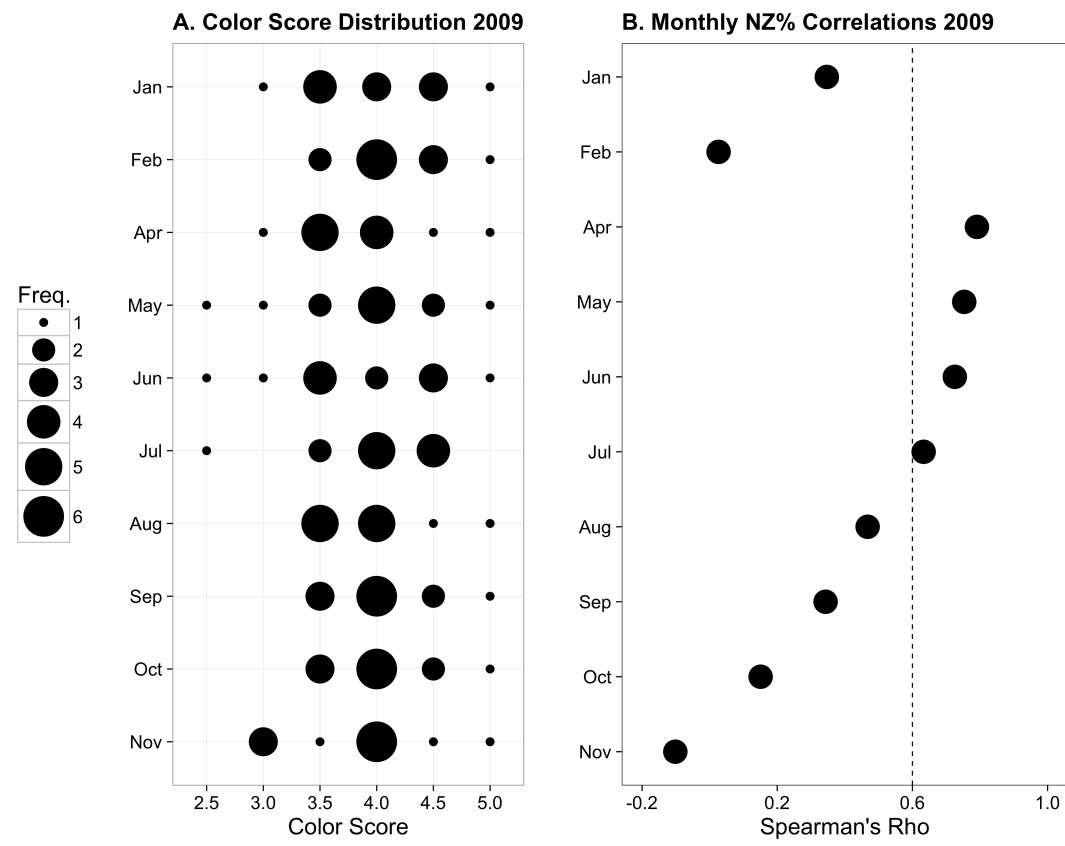

**Figure 3** **Previously published data from monthly monitoring of 12 *Palythoa tuberculosa* colonies from similar sites in Okinawa Island, Japan.** (A) Color score distribution by month. (B) Correlations (Spearman's rho) between colony color score and normal zooxanthellae percent by month. The dotted line indicates the region above which rho values are statistically significant for $\alpha = 0.05$. Modified from *Hibino et al. (2013)* with permission.

## Correlations among variables

Mean symbiont morphological condition (NZ%) ranged from 71.0 to 86.8% normal (Fig. 2B). Mean symbiont cell density ranged from $0.18 \times 10^6$ to $1.96 \times 10^6$ cells cm$^{-3}$ (Fig. 2C). Chlorophyll *a* concentration ranged from 2.4 to 18.2 µg cm$^{-3}$ (Fig. 2D). Color score trended toward a slightly negative but statistically insignificant correlation with chlorophyll *a* ($\rho = -0.472$, $p_{adj} = 0.215$), and had no relationship with symbiont morphological condition ($\rho = -0.027$, $p_{adj} = 1$) or cell density ($\rho = -0.030$, $p_{adj} = 1$).

## Comparisons between color fingerprinting and recurrent monitoring time series data

From January to November 2009, the color score distributions for 12 repeatedly sampled *P. tuberculosa* colonies were fairly consistent, with most scores lying in the range from 3.5 to 5.0, the exception being summer months, when some colonies scored as low as 2.5 (Fig. 3A). Color score and symbiont morphological condition (NZ%) were only significantly positively correlated in summer months (Fig. 3B), for example, in May ($\rho = 0.753$, $p_{adj} = 0.004$; Fig. 4A) and June ($\rho = 0.726$, $p_{adj} = 0.009$; Fig. 4B). When

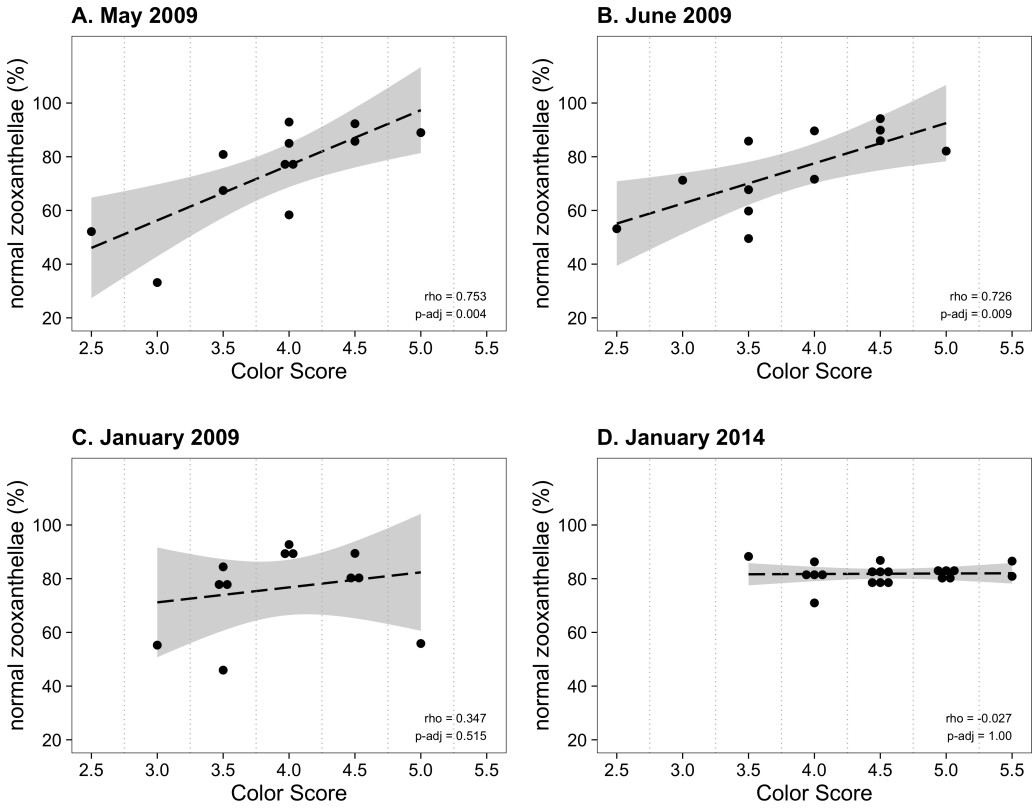

**Figure 4** **Normal zooxanthellae percent values segregated by color score for colonies of *Palythoa tuberculosa*.** (A) May 2009, (B) June 2009, (C) January 2009, and (D) January 2014. (A) and (B) represent summer months with large and significant correlation coefficients (Spearman's rho). (C) and (D) represent winter months with little correlation. Note that (A–C) represent the same set of repeatedly sampled colonies, whereas (D) represents a unique set of colonies. Dashed black lines indicate linear regression fits. Gray shading indicates 95% confidence intervals. Rho values are Spearman correlation coefficients. Significance was assessed at $\alpha = 0.05$. (A–C) modified from *Hibino et al. (2013)* with permission.

January 2009 recurrent monitoring data were compared to the color fingerprinting data from January 2014 (Fig. 4C vs. Fig. 4D), both distributions were similar, with most colonies exhibiting healthy color and NZ% ranges and no correlation between metrics.

## DISCUSSION

Reef managers are faced with many options when deciding how to allocate limited resources to enhance local conservation efforts. The 'CoralWatch Coral Health Chart' was developed as a low-cost, non-invasive technique for assessing reef health (*Siebeck et al., 2006*). Recurrently monitoring individual colonies and color fingerprinting random colonies have both been proposed as effective citizen science methods (*Siebeck, Logan & Marshall, 2008*). Using the potential 'early-indicator' zoantharian *Palythoa tuberculosa* from Okinawan reefs, we found that both techniques complement each other and work best when used in conjunction, in line with the recommendations from CoralWatch. However, zoantharian-based color monitoring programs face unique problems that may restrict their utility.

It can be difficult to ensure data validity in any citizen science program (*Foster-Smith & Evans, 2003*), but the issue is compounded for color surveys in marine environments. Human color perception is inherently variable and affected by altered spectral properties underwater (*Neitz & Jacobs, 1986*; *Winters et al., 2009*). While the color cards are designed to reduce subjectivity, there are also issues regarding natural color variation among cnidarian symbioses. For example, the number of symbiont cells in a given colony increases and decreases seasonally (*Fitt et al., 2001*), and coral colonies that appear white are not necessarily bleached or compromised (*Cruz et al., 2015*). Both sources of variation presented challenges in this study.

Based on low between-observer error in color scores, volunteers were successfully trained to moderate skill levels quickly. The high correlation among individuals (Spearman's $\rho = 0.633$) despite a low reliability statistic (Krippendorf's $\alpha = 0.555$) meant observers consistently ranked colonies in similar order, though often failed to assign exactly the same scores. Despite absolute score discrepancies, deviations remained low and even the inclusion of a difficult-to-score colony (O7) did not unduly influence the reliability or correlation measures. One major advantage of color fingerprinting over recurrent monitoring is that more volunteers can typically be included, so losses in color accuracy due to the involvement of many non-experts can be offset by averaging observations. When designing volunteer surveys, the best practice appears to be to maximize the number of colonies that are photographed, and afterwards to have as many individuals as possible (at least 3–4) score every image independently. This improves confidence that the rounded mean color score reflects the true color.

However, accurate color measurement does not matter if color is not indicative of symbiosis quality. Based on our color fingerprinting data from January 2014, we found that color scores did not predict the values of more invasive metrics such as symbiont morphological condition, cell density, or chlorophyll *a* content (Fig. 2). Only observations from relatively stressful warm summer months resulted in high correlations among color and symbiont morphological condition (Figs. 3B and 4), suggesting the utility of the color score approach depends upon the presence of compromised colonies with low symbiosis quality. During the cool subtropical winter, most *P. tuberculosa* colonies were healthy, having high percentages of 'normal' (non-degraded) zooxanthellae, high symbiont cell densities, and generally high color scores. As noted in the original description of the color card technique, the relationship between color and other metrics is weakest among darker, healthier colonies (*Siebeck et al., 2006*). This points to a general issue with color data; while it can be good for detecting bleached colonies (but see *Cruz et al., 2015*), it is potentially uninformative for healthy colonies above certain thresholds, which are likely species- or location-specific (*Kemp et al., 2006*). A large degree of natural color variation makes *P. tuberculosa* a problematic focal species for bleaching monitoring.

Our results suggest a single color fingerprinting data set does not appear to be useful on its own, unless the goal is to compare several reefs within a management area simultaneously (e.g., *Siebeck, Logan & Marshall, 2008*). One isolated snapshot can be difficult to interpret without a baseline for comparison, especially during winter months. Our combined analyses of Okinawan *P. tuberculosa* colonies from 2009 to 2014 indicates that data generated from
color fingerprinting reflect a major trend from recurrent monitoring; namely that under winter conditions colonies are generally healthier and darker (Fig. 3A). Hence, the two methods can complement each other. For example, using the *P. tuberculosa* recurrent monitoring results as a predictor, we anticipate that color fingerprinting anywhere from April to August will include a broad range of colors and health conditions. A deviation from this trend (such as an overrepresentation of lighter colonies) may signify bleaching conditions on a reef. However, the exact expectation of which months will be useful to survey will vary by species and reef, which is why local baseline data are so crucial. If feasible, we encourage reef managers to adopt a combined approach to color observation: first having specialists track tagged colonies over the course of a year, then having volunteers fingerprint during critical time periods.

This particular citizen science effort was structured around a university field course, presenting some unique difficulties. The training period was brief (two days in a classroom setting), and while students appeared consistent in their scoring of reference photographs, more training might have improved the reliability of their data. We were also limited in the time available for sample collection. We sampled only 20 colonies, falling far short of the standard set by other studies (e.g., 100 colonies; *Siebeck, Logan & Marshall, 2008*). Finally, the course was scheduled for winter, though it would have been more ecologically relevant to sample in summer. These limitations point to several avenues for further development of zoantharian color monitoring programs. Future work could test the impact of training duration on color score reliability, or examine bleaching summers vs. non-bleaching summers. Photochemical interrogation via pulse-amplitude modulated (PAM) fluorometry could be used to determine whether zoantharian symbioses with light coloration are truly compromised or simply extreme points on the spectrum of natural color variation. Because Okinawa Island has been spared from major bleaching in 2009 and 2014, it was impossible to test with this particular data set whether *P. tuberculosa* functions as an "early indicator" species for the bleaching of scleractinian corals in the area, but continued monitoring should provide the observations necessary to establish such a link if it exists. Such work is needed to confirm the utility of a citizen science approach to monitoring zoantharian color.

## ACKNOWLEDGEMENTS

We would like to thank the 12 participating members of the 2013–2014 "Kaiyo Seibutsu Seisan" Practical Course VII at the University of the Ryukyus and Taku Ohara for their hard work and enthusiasm.

### Funding

This work was supported by the Japan Society for the Promotion of Science and the International Research Hub Project for Climate Change and Coral Reef/Island Dynamics at the University of the Ryukyus. The funders had no role in study design, data collection and analysis, decision to publish, or preparation of the manuscript.

### Grant Disclosures

The following grant information was disclosed by the authors:
Japan Society for the Promotion of Science.
International Research Hub Project.

### Competing Interests

James Davis Reimer serves as an Academic Editor for PeerJ.

### Author Contributions

- John Everett Parkinson conceived and designed the experiments, analyzed the data, wrote the paper, prepared figures and/or tables, reviewed drafts of the paper.
- Sung-Yin Yang performed the experiments, contributed reagents/materials/analysis tools, reviewed drafts of the paper.
- Iori Kawamura and Gordon Byron performed the experiments, reviewed drafts of the paper, assisted in field course instruction.
- Peter Alan Todd conceived and designed the experiments, reviewed drafts of the paper.
- James Davis Reimer conceived and designed the experiments, performed the experiments, analyzed the data, wrote the paper, prepared figures and/or tables, reviewed drafts of the paper, instructed field course.

### Data Availability

The data and code are provided as Data S1.

### Supplemental Information

Supplemental information for this article can be found online at http://dx.doi.org/10.7717/peerj.1815#supplemental-information.

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
