# Peer review of "A citizen science approach to monitoring bleaching in the zoantharian Palythoa tuberculosa"

_PeerJ, doi:10.7717/peerj.1815_

## Round 0.1 · original submission · Major Revisions

All reviewers find the strategy to use grassroots efforts for monitoring a laudable effort, especially given the current relevance of cnidarian bleaching. Reviewer #3 raises important concerns regarding the skills that can be reliably developed by naive participants after only 2 days of training and as such the validity of the approach is weakened. The same reviewer also raises an important concern, which is time of sampling in winter months. I understand that possibly had to do with the scheduling of the undergraduate course. I recommend you tone down the assertions about the universality of this method until is better developed. At the same time encouraging other groups elsewhere to try low cost "citizen science" as Reviewer #3 mentions is an important aspect of this contribution and therefore I consider worth resubmitting. I encourage you to address all the comments raised by the reviewers to improve your manuscript.

Reviewer 1 ·

Basic reporting

The manuscript highlights the importance of the identification of early indicator species to detect or anticipate bleaching events when assessing phenotypic changes. It also highlights the importance of involving the non-scientist community into conservation efforts and awareness of environmental problems related to coral reefs. It is well structured and written. It has a clear introduction and background knowledge. Though, there are important limitations resulting from the experimental design as to conclude that the species and the techniques used (fingerprinting in midwinter) could be potential indicators of coral bleaching. Observations and concerns are stated in the Comments for the author section

Experimental design

The experimental design involves simple tools and techniques to determine changes in colony colors and its correlation with other more complex health metrics. It tested the reliability of the data gathered by observers. The combined analysis was a good resource to recall the seasonal variability of health conditions in this species. The little predictive power of the other variables in midwinter should be expected considering the background knowledge

Validity of the findings

Very useful approach for monitoring reefs at a local scale and detect benthic organisms’ health-condition with simplified tools and techniques. Further assessments and analysis on thermal stress conditions should be done to validate the probability of this species as potential coral bleaching indicator

Additional comments

Lns 54-59. There are more recent reports and research on bleaching that may be included in the background.
Ln 71. Methodology from CoralWatch does not require “trained divers photograph”. Siebeck et al. (2006) included pictures in their methodology to validate the colour reference card.
Ln 75. Hibino et al. (2013) considered the “morphological condition”. The potential relation of the “symbiont condition” with the color of the colony is not described.
Ln 117. Why “doubling the resolution of the original scale”?
Ln 121. “Cell condition” is a very ambiguous term. “Morphological condition”, as in Reimer et al. (2007), is more specific. Do the category “Normal zooxanthellae NZ” corresponds to “non-degraded zooxanthellae” (as in Ln 251)?
Ln 130. Siebeck et al. (2008) randomly selected 100 corals while fingerprinting the reef condition at Heron Island Reef. Taking this study as a reference point, do you think ten colonies are an appropriate sample size for “fingerprinting” the reef condition at the two locations assessed?
Ln 209. According to the methodology, symbiont and chlorophyll concentration were calculated over a volume (cm3), but results are expressed in area units (cm2).
Ln 262. Symbiosis health?
Lns 261-266. Check wording. It is not clear why the relationship between color and other metrics is different in both cases.

Reviewer 2 ·

Basic reporting

In general, the manuscript follows PeerJ policies. The introduction states that the field of knowledge is coral bleaching, however it should be bleaching due to the use of zoantharian species. As a suggestion, the introduction as well as the title should be modified to bleaching or cnidarian bleaching, instead of coral bleaching. If these changes take place, the references should change. Present introduction literature is relevant.

The structure of the manuscript follows the PeerJ templates. All figures are relevant, of sufficient resolution and properly described.

Experimental design

The article originality is the use of student volunteers to assess, after training, the bleaching status of the zoantharian. This is based on the use of "CoralWatch Coral Health Chart" and the results obtained by Hibino et al. (2015).

This manuscript presents an interesting research, however there are some parts that need to be rewritten, such as the objective and hypothesis (research question). These are not clearly stated. It is important for the authors to rewrite the paragraph that describes the scope of the study focusing it on cnidarian (zoantharian) bleaching.

The description of the methods used is clear for other researchers to reproduce them. As stated by the authors, the zoantharian species is free from legal protections.

Validity of the findings

The experimental design accomplish the PeerJ requirements, however the colour scoring data taken in 2014 is limited. The later, explain the results obtained (little predictive power) and the necessity for using the raw data from Hibino et al. (2013). The use of both data set as well as the data obtained in the laboratory support the conclusions regarding the zoantharian bleaching, however it does not to coral bleaching.

Additional comments

As mentioned before, the manuscript has to be modified and focused in cnidarian bleaching. Additionally, in the discussion the practical considerations should be extended.

Reviewer 3 ·

Basic reporting

The article is generally well written and organized to the standards of Peer J. This works is attempting to use “citizen science” approaches and methodologies combined with empirical data to make inferences about anthozoan bleaching.

Introduction is mostly complete. I do not believe that the term ‘fingerprint’ should be used throughout the manuscript to describe the qualitative assessment (from citizen scientists) of the bleaching status of the zoanthids. The general term ‘fingerprinting’ includes methodologies that are well documented using genetic markers (DNA fingerprinting) or spectral analyses (Ecological fingerprinting). Both of these techniques use rigorous, and quantitative approaches that are widely accepted. I do not believe the proposed techniques used to ‘fingerprint’ the biological status of zoanthids in this manuscript are of the same standard.

The Introduction and Discussion should consider the findings of: Cruz et al. Marine Biology 2015. White but not bleached: photophysiological evidence from white Montastraea cavernosa reveals potential overestimations of coral bleaching.

Experimental design

See comments in “Basic Reporting” referring to the usage of the term ‘fingerprint’.

Portions of this research are done quite rigorously and I find the results relevant and meaningful. For example, using Symbiodinium densities and Chl a measurments to compare colour scores is well done and demonstrates the amount of variation that can be associated with the “colour” of a anthozoan and the colour doesn’t represent the health status of the organisms (during most conditions). Suggest incorporating Cruz et al. 2015 throughout the manuscript.

Furthermore, zoanthids are somewhat unique in that they incorporate and accrete large amounts of sediment (carbonate materials) into their tissue layers. How this the amount of particulates no doubt will influence light scattering and overall color of the organism. Therefore, overall color of zoanthids may not very informative on the overall physiological status of the organism; thus, zoanthids may not be very good indicator species for overall coral bleaching.

The use of visually inspecting symbiont cell condition does not seem very informative. What exactly is being assessed and quantified? I am very skeptical of what cellular and biochemical information can be measured (without advanced staining and cellular probes) via microscopy. At the minima photographs of what is deemed “healthy” or “unhealthy” Symbiodinium should be included as a figure. However, preservation techniques, isolation of Symbiodinium, and storage of samples will all influence how Symbiodinium cells “look”.

Although I applaud incorporating undergraduates into a research and environmental monitoring program e.g. “Twelve undergraduate students from the University of the Ryukyus were trained as part of a two-day practical course using images of Palythoa tuberculosa and Symbiodinium sp. collected previously as part of a year-long monitoring project”. I do not see how they can become familiar enough with what healthy looking Symbiodinium cells are to accurately collect empirical data. The same is true with the color score tests. Two days in a class room setting seems to be stretching the capabilities of the analyses.

Typically how humans perceive the color of anthozoan has very little to do with the health status of the organism, with the exception of very bleached organisms. This work documents the variation associated with visual classifications. There seems to be some disagreement between some of the conclusions of this work. For example, “We found that P. tuberculosa colour score, although highly correlated among observers, provided little predictive power for the other variables. This was likely due to the overall healthy condition of the reef in midwinter. Nevertheless, when used in conjunction with baseline monitoring data, fingerprinting is an easy and robust means of assessing reef health as part of a larger conservation strategy.” These statements seem to contradict themselves.

Additionally, comparing winter, non-stressful conditions, to other time points doesn’t seem very ecological relevant. A better strategy would be compare health summer conditions to stressful conditions. This would allow for more similar photoacclimation and seasonal acclimation of the Symbiodinium and zoanthids and comparisons to “bleached” conditions would be more accurate.

An additional point of contridction was at Ln 242-244 where it is stated that “One major advantage of fingerprinting over monitoring individuals is that more volunteers can typically be included, sp losses in accuracy due to the involvement of many non-experts can be offset by averaging observations.” However, just in the next paragraph they state “Based on our fingerprinting data from January 2014, we found that colour scores did not predict more invasive metric values such as …”

Validity of the findings

Due to some of the problems with the experimental design listed above (i.e. 2-days of student training to asses dinoflagellate “health” and coral color) and the comparison of winter acclimated versus summer “stressed” zoanthid colonies, I think many of the predictions are not sound. Furthermore, the variation in color scores alone should indicate that this way of monitoring zoanthids is likely not very valid. I like the attempt of incorporating citizen science into environmental monitoring, however, I believe that this paper does a poor job justifying this approach is a sufficient tool to do such.

---

## Round 0.2 · accepted · Accept

I am sorry in the delay in getting this decision back to you, but I had some catching up to do before I could make an informed decision. After reading through the original reviews and your revision with tracked changes and the referees appraisal of the revision, I am satisfied that you have addressed the major concerns of all three reviewers. Only two of the referees responded to the request to evaluate the revision, so we will move forward without additional comments from the third referee. In terms of the revision, reviewer #1 comments that you make some assumptions based on comparisons with the scleractinian literature that you may wish to discuss and refocus your discussion more on zoanthids, but I leave that decision to you. In reading your revised manuscript, I personally feel that the results are sufficiently qualified and the directions for future refinement of the technique are clearly spelled out for any reader, and that you do not overstate the data without appropriate caveats. Thus, if you wish to modify the text slightly as suggested by the referee, you are welcome to do so, but I find myself in agreement with the second referee, and do not feel that it is absolutely necessary to move forward with acceptance of this manuscript.

Reviewer 1 ·

Basic reporting

In general, it is a well written and organized manuscript, and follows PeerJ standars. I congratulate the authors for their approach on including citizen science into environmental monitoring, although this brings unique difficulties as clearly stated. The introduction included appropriate background knowledge and indicates the importance of the species and the techniques used. Methods and results were sufficient and clear, according to the logistical issues.

I consider that the discussion could be improved and focused more on zoantharian bleaching, than on coral bleaching. Even if both taxa respond similarly to heat stress, symbiotic scleractinians and zoantharians have distinctive morphological and anatomical characteristics that should be considered after their important role on the spectroscopic properties of the holobiont.

Experimental design

The research question is well defined and the methods are described with sufficient detail for other researchers to reproduce them. Some of the findings are not entirely new and have been reported in previous studies (e.g. the little predictive power of the color score for other variables in midwinter); though the research fits the scope of the journal.

Validity of the findings

Some assumptions are based on scleractinian corals literature and the extension to anthozoans should be carefully stated. For example: “Average colony scores in this study ranged from 3.50 to 5.63, corresponding to a rounded range of 3.5 to 5.5. This span is typically associated with healthy colonies (Siebeck et al., 2006; Siebeck et al., 2008)”. The anatomical and morphological differences between anthozoans and scleractinians should be further discussed since they may determine differences in the color perception, especially if proposing this species as an early indicator for scleractinian corals bleaching.

Reviewer 2 ·

Basic reporting

The authors address all the comments raised by the reviewers and modified the manuscript following these comments. The latest manuscript, as in its original version, is well written and follows all PeerJ standards. Additionally, the new title fits better the hypothesis and the citizen science approach.

Experimental design

Although the experimental design has some issues, as stated in some of the comments from the reviewers, the use of students, simple techniques and more complex data is original. The changes made by the authors in the manuscript clarify some of the concerns bring up by the reviewers.

Validity of the findings

As stated previously this findings of this study are interesting and with the changes made by the authors the manuscript has improved. Some of the concerns of the reviewers regarding the limited time of the training and the time of the sampling (winter) are properly address by the authors in this new version of the manuscript.